# Experimental Identification of Cross-Reacting IgG Hotspots to Predict Existing Immunity Evasion of SARS-CoV-2 Variants by a New Biotechnological Application of Phage Display

**DOI:** 10.3390/v16010058

**Published:** 2023-12-29

**Authors:** Marek Adam Harhala, Katarzyna Gembara, Krzysztof Baniecki, Aleksandra Pikies, Artur Nahorecki, Natalia Jędruchniewicz, Zuzanna Kaźmierczak, Izabela Rybicka, Tomasz Klimek, Wojciech Witkiewicz, Kamil Barczyk, Marlena Kłak, Krystyna Dąbrowska

**Affiliations:** 1Research and Development Center, Regional Specialist Hospital in Wrocław, Kamieńskiego 73a St., 51-124 Wrocław, Poland; marek.harhala@hirszfeld.pl (M.A.H.); katarzyna.gembara@hirszfeld.pl (K.G.); natalia.jedruchniewicz@wssk.wroc.pl (N.J.); zuzanna.kazmierczak@hirszfeld.pl (Z.K.); tomasz.klimek@wssk.wroc.pl (T.K.); wojciech.witkiewicz@wssk.wroc.pl (W.W.); marlena.klak@wssk.wroc.pl (M.K.); 2Hirszfeld Institute of Immunology and Experimental Therapy, Weigla 12 St., 53-114 Wrocław, Poland; izabela.rybicka@hirszfeld.pl; 3Healthcare Centre in Bolesławiec, Jeleniogórska 4, 59-700 Bolesławiec, Poland; krzysztofbaniecki@gmail.com (K.B.); aleksandrapikies@gmail.com (A.P.); anahorecki@zozbol.eu (A.N.); sekretariat@szpitalboleslawiec.pl (K.B.)

**Keywords:** SARS-CoV-2, COVID-19, antibodies, IgG, epitopes, phage display, serological profiling

## Abstract

Multiple pathogens are competing against the human immune response, leading to outbreaks that are increasingly difficult to control. For example, the SARS-CoV-2 virus continually evolves, giving rise to new variants. The ability to evade the immune system is a crucial factor contributing to the spread of these variants within the human population. With the continuous emergence of new variants, it is challenging to comprehend all the possible combinations of previous infections, various vaccination types, and potential exposure to new variants in an individual patient. Rather than conducting variant-to-variant comparisons, an efficient approach could involve identifying key protein regions associated with the immune evasion of existing immunity against the virus. In this study, we propose a new biotechnological application of bacteriophages, the phage display platform for experimental identification of regions (linear epitopes) that may function as cross-reacting IgG hotspots in SARS-CoV-2 structural proteins. A total of 34,949 epitopes derived from genomes of all SARS-CoV-2 variants deposited prior to our library design were tested in a single assay. Cross-reacting IgG hotspots are protein regions frequently recognized by cross-reacting antibodies in many variants. The assay facilitated the one-step identification of immunogenic regions of proteins that effectively induced specific IgG in SARS-CoV-2-infected patients. We identified four regions demonstrating both significant immunogenicity and the activity of a cross-reacting IgG hotspot in protein S (located at NTD, RBD, HR1, and HR2/TM domains) and two such regions in protein N (at 197–280 and 358–419 aa positions). This novel method for identifying cross-reacting IgG hotspots holds promise for informing vaccine design and serological diagnostics for COVID-19 and other infectious diseases.

## 1. Introduction

Efficient immunological protection in COVID-19 is currently the major factor shaping the global perspective on the growth or recession of further pandemic waves. The SARS-CoV-2 virus has constantly been producing a multitude of variants; many of these variants, such as Delta or Omicron, triggered pandemic waves obviously using evasion of existing immunity against the virus to spread within the global population [1,2]. Importantly, antigenic drift, driven by the pressure from specific immunity exerted on the virus, significantly contributes to the selection of new variants able to evade immunological control [3]. For these reasons, the hope for efficient control and suppression of the pandemic simply through global vaccinations and immunological memory from previous infections has been tempered.

Virus-specific antibodies are, in fact, a pool of antibodies targeting different structural proteins. Structural proteins of SARS-CoV-2 include nucleocapsid protein (protein N), spike protein (protein S), envelope protein (protein E), and membrane protein (protein M) [4]. The most important target for the immune response is protein S, due to the virus-blocking potential of antibodies targeting this protein. This protein mediates binding to the ACE2 receptor; thus, antibodies targeting the S protein may neutralize the virus. The S protein is relatively large, 1273 amino acids (aa) long (acc. no.: YP_009724390.1), and it consists of two major domains: S1 and S2. The S1 subunit function is binding to receptors in human cells. Its major parts are the N-terminal domain (NTD) and receptor binding domain (RBD). The S2 subunit, in turn, is engaged in the viral and host cell membranes’ fusion. S2 consists of the cytoplasmic tail (CT), transmembrane domain (TM), heptad repeat 1 (HR1), heptad repeat 2 (HR2), fusion peptide (FP), central helix (CH), and connector domain (CD). Also, the cleavage site S1/S2 plays a key role in virus entry to a human host cell. Linear epitopes on the S protein have been demonstrated to elicit neutralizing antibodies in COVID-19 patients [5]. The N protein, in turn, can be a target for serological diagnostics, and at the beginning of pandemics, it was used in the majority of serological tests. Later, it was replaced by protein S, due to the direct indication of S-targeting responses on the neutralization potential of antibodies, and also due to potential problems with specific response induction in asymptomatic infections. However, the relatively conserved protein N is still being used to confirm SARS-CoV-2-related etiology of already cleared infections, since the N protein is unique for SARS-CoV-2, highly immunogenic, and not included in anti-COVID-19 vaccines [6]. Two other proteins, M and E, seem to have lower significance as potential targets for specific antibodies. However, one cannot exclude the role of these antibodies in virus inhibition.

After two years of the pandemic and the continuous emergence of new variants, it is challenging to thoroughly understand all the possible combinations of previous infections, various vaccination types, and potential exposure to new variants in an individual patient. These combinations further generate myriads of possibilities at the population level. Thus, instead of variant-to-variant comparisons, the identification of key protein regions that are linked to the immune evasion of existing immunity against the virus could be more helpful for predictions on pandemic potential in new SARS-CoV-2 variants. 

Here, we validate a new method for the experimental identification of regions that can play a role in *cross-reacting IgG hotspots* in SARS-CoV-2 structural proteins. The *cross-reacting IgG hotspots* have been identified by a high-throughput assay of cross-reactivity of antibodies elicited by SARS-CoV-2, based on the modified VirScan technology [7]. This approach employs a phage display library of epitopes, and it allows for significant extension of the most popular method for B-cell epitope identification (by microarrays) and for massive testing of a multitude of epitopes. We affirm the validity of identifying immunogenicity regions in both our current protocol and previous methodologies as it constitutes the core application of all VirScan-derived assays. By extending the scope of the library, experimental assay, and calculations to encompass multivariant analysis, we have introduced a powerful new tool for predictions grounded in empirically observed interactions. Consequently, we propose a combined approach involving the identification of immunogenicity regions and the pinpointing of cross-reacting IgG hotspots. This synergistic strategy promises to yield more valuable insights into the immunogenicity of the virus (or other pathogens). Here, 34,949 oligopeptides have been tested in a single assay, thus representing all variants of SARS-CoV-2 available in databases at the moment of the library creation (Appendix A).

## 2. Materials and Methods

### 2.1. Serum Donors

Blood serum was collected from patients hospitalized due to COVID-19: PCR-confirmed infection with SARS-CoV-2 (at least 1 month after infection), non-vaccinated, over 18 years old, hospitalized in COVID-19 ward in Healthcare Centre Bolesławiec, Poland from 27 January 2021 to 26 March 2022 (alpha variant: n = 15, delta variant: n = 8). 

### 2.2. Bioethics Statements

The study was conducted in accordance with the principles of the Declaration of Helsinki. The research was approved by the local Bioethical Commission of the Regional Specialist Hospital in Wroclaw (approval number: KB/02/2020, policy No. COR193657). During the individual interview, all information about the study was provided and written consent was obtained from each participant. The written consent was accepted by the local Bioethical Commission of the Regional Specialist Hospital in Wroclaw (approval number: KB/02/2020).

### 2.3. Blood Samples

Blood samples were collected in test tubes (BD SST II Advance), left to clot for 1 h at room temperature (RT), and serum was separated from the clot by centrifugation (15 min, 2000 g, RT) and then stored at −20 °C for further use.

### 2.4. Identification of SARS-CoV-2 Variants Infecting Investigated Patients

In selected hospitalized patients, SARS-CoV-2 variants were identified from biological material collected as nasopharyngeal swabs. The material was processed for standard RNA isolation and applied to the targeted sequencing approach with the Ion AmpliSeq SARS-CoV-2 Insight Research Assay (Thermo Fisher Scientific, Waltham, MA, USA, according to the manufacturer’s instructions). Briefly, patients were sampled by nasopharyngeal swabbing with sterile medical swabs (Chemivet, Olsztyn, Polska), viral RNA was isolated on silica spin columns (QIAquick), and the viral load was quantified by the qPCR standard diagnostic test (primers 5′-ACAGGTACGTTA ATAGTTAATAGCGT-3′, 5′-ATATTGCAGCAGTACGCACACA-3′, 5′-GGG GAA CTT CTC CTG CTA GAA-3′, 5′-CAG ACA TTT TGC TCT CAA GCT G-3′). Ct values were used to design library preparation parameters, which included standard retrotranscription. Library preparation and sequencing chip loading were completed on Ion Chef. Sequencing was completed on Ion GeneStudio S5 and analyzed with the software provided by the manufacturer (Thermo Fisher Scientific, Waltham, USA). Sequences of sufficient quality were uploaded to the public database GISAID (https://www.gisaid.org/ accessed on 17 December 2021).

### 2.5. Epitope Analysis

Identification of oligopeptides interacting with specific IgGs was performed in accordance with the modified protocol published by Xu et al. [7] and adapted for our research using coding sequences of the investigated SARS-CoV-2 variants as the source for library design [7]. Part of a library representing protein variants was created from proteomes of SARS-CoV-2 variants downloaded on 7 April 2021 from the Identical Protein Groups Database, National Center for Biotechnology Information. Another part of a library representing the reference proteome of the SARS-CoV-2 virus (prod. ID: UP000464024) was created from the Proteome Database at UniProt Proteome resources, and a set of oligopeptides was created by single and triple alanine substitution, similar to the original procedure [7]. Alanine substitutions have been established in VirScan technology as the method for epitope disruption. Specifically, alanine is substituted for each amino acid in the investigated sequence of the epitope, both individually and in triplicates. In the case of original sequences containing alanines, the substitution is completed with glycine. These substitutions are employed to identify specific amino acids involved in epitope–antibody interactions. Disruption of the interaction occurs when a key amino acid is substituted.

Variants of each investigated reference protein were aligned with the Clustal Omega software v. 1.2.4 [8,9,10]. Each alignment and protein library consisting of reference proteomes was virtually cut into 56 aa long fragments tailing through protein sequences, starting every 28 aa from the first amino acid. The peptide library, consisting of protein variants in the Identical Protein Group, was cleared from all incomplete sequences (containing missing fragments or undetermined amino acids). If a specific protein showed a gap in the alignment, oligopeptides covering this place were shorter than 56 aa. The start position of each oligopeptide in the reference proteins N, S, M, and E of SARS-CoV-2 is presented in Appendix A. The resulting library contained 34,949 epitopes.

Sequences of all oligopeptides were reverse-translated into DNA sequences using codons optimized for expression in *E. coli*. The oligopeptide library was synthesized using the SurePrint technology for nucleotide printing (Agilent, Santa Clara, USA). These oligonucleotides were used to create a phage display library using the T7Select 415-1 Cloning Kit (Merck Millipore, Burlington, USA). Immunoprecipitation of the library was performed in accordance with a previously published protocol [7,11]. Briefly, the phage library was amplified in a standard culture as described in the manufacturer’s manual and then purified by hollow fiber dialysis against a Phage Extraction Buffer (20 mM Tris-HCl, 100 mM NaCl, 6 mM MgSO_4_, pH 8.0). All plastic containers (96-well plates) used for immunoprecipitation were prepared by blocking with 3% Bovine Serum Albumin (BSA) in a TBST buffer overnight on a rotator (50 rpm, 4 °C). A sample representing an average of 10^5^ copies of each clone in 250 µL was mixed with 1 µL of human serum (two technical replicates were applied) and incubated overnight at 4 °C with rotation (50 rpm). A 20 μL aliquot of a 1:1 mixture of Protein A and Protein G Dynabeads (Invitrogen, Waltham, USA) was added and incubated for 4 h at 4 °C with rotation (50 rpm). The liquid in all wells was separated from Dynabeads on a magnetic stand and removed. Beads were washed 5 times with 280 µL of a wash buffer (50 mM Tris-HCl, pH 7.5, 150 mM NaCl, 0.1% Tween-20) and the beads were resuspended in 60 µL of water to elute the immunoprecipitated bacteriophages from the beads.

The immunoprecipitated part of the library (for each patient-derived sample), as well as 20 samples representing the library before immunoprecipitation (input samples), was then used for amplification of the insert region according to the manufacturer’s instructions with a Phusion Blood Direct PCR Kit (Thermo Fisher Scientific, Waltham, USA) (primers 5′-TCGTCGGCAGCGTCAGATGTGTATAAGAGACAGGATCCGAATTCTTCTTCTTCT-3′, 5′-GTCTCGTGGGCTCGGAGATGTGTATAAGAGACAGTTAACTAGTTACTCGATGC GG-3′). A second round of PCR was carried out with the IDT for Illumina UD indexes—set: A, B, C (Illumina Corp., San Diego, USA) to add adapter tags. The products were purified using AMPure XP magnetic beads (Beckman Coulter, Brea, USA). Sequencing of the amplicons was completed using Illumina next-generation sequencing (NGS) technology (Genomed, Warszawa, Poland). A full list of oligopeptides in the tested library (cloned) is given in Appendix A.

### 2.6. Sequencing Data Analysis

Sequenced amplicons were mapped to the original nucleotide library sequences by the bowtie2 software v. 2.3.4.1, similarly as described by Xu et al. [7,12]. NGS sequencing reads (after removal of sequences added in PCR) were mapped to the full list of oligonucleotides as designed for the library synthesis as indexes (options: end-to-end mode, ‘-q -5 9 --no-unal --no-hd --no-sq --ignore-quals --mp 3 --rdg 150,100 --rfg 150,100 --score-min L,-0.6,-0.6′) [12,13]. The number of hits that were mapped to each reference sequence (only the highest score for each read) was counted (*count*, *c*).

The signal in each sample was calculated/normalized according to Formula (1):(1)$$s_{ijm}=\frac{c_{ijm}}{\sum_{i\in I}c_{ijm}}$$

*s*—a signal of the *i*-th sequence in the *j*-th serum sample and *m*-th technical replicate.

*c*—count, a number of readings mapped to the *i*-th sequence in the *m*-th technical replicate of the *j*-th serum sample.

*I*—set of all reference sequences (used as indexes in mapping by the bowtie2 software).

Signals in each *input sample* used as negative control samples (n = 20, amplified and purified phage library sequenced before immunoprecipitation) were calculated. In the input samples, the average log_10_ of a signal and its standard deviation (SD) for each tested sequence were used as the *control signal* for later calculations of *p*-values. As for the tested samples, the log_10_(signal) was determined for each detected sequence (*count* > 0). In order to avoid false positive results, we set the signal of undetected oligonucleotides (*s*) in a tested sample to the minimum measured for this sample value and set its standard deviation to the average value of SD in the input samples.

Each log_10_(signal) in the tested samples was compared to the *control signal* of the same sequence in the input samples, and the *p*-value was calculated (assuming normal distribution). Only signals detected with a non-zero count and *p*-value < 0.05 in both technical replicates of a sample were recognized as *positive* (significantly enriched). *Relative signal* (enrichment): an average signal in technical replicates of a sample divided by the average signal in ‘input samples’ of the same sequence resulting in a signal ratio (*relative signal*). The above procedure was tested on the input samples as controls and proved to yield less than a 5% chance for a false *positive* (significantly enriched) sequence in each sample.

### 2.7. Immunogenicity and Cross-Reacting IgG Hot-Spot Determination—Statistical Model

Determination of immunogenicity and cross-reacting IgG hot-spot was performed by adopting binomial distribution. This distribution (provided with *chance* and *size*) allows one to calculate the chance of randomly observing a specific amount of positive outcomes in a series of trials. The assumption is that every patient has the same chance of recognizing each measured oligopeptide and its variant separately.

For each group (Alpha/Delta, N/S-protein) in both experiments (immunogenicity research and hot-spot analysis) we used the same protocol separately. First, we calculated *a chance* by dividing the sum of all detected (positive) oligopeptides in each patient in a group by all tested patients and all tested oligopeptides/variants. *Size* is the amount of patients in a researched group (immunogenicity) or the number of tested variants of a fragment in question (cross-reacting hotspots). Then, in R language, we calculated the chance for a random occurrence of an observed number of positive results for each tested protein fragment. Fragments whose chance for a random occurrence of a measured result is below 0.05 are indicated in the figures with one star and results below 0.001 are indicated with double stars. Exact *p*-values are provided (Appendix A) along with the graphical comparison of the statistical model used and observed data (Appendix A). 

## 3. Results

The immunoreactivity of specified regions (oligopeptides) within SARS-CoV-2 proteins was assessed using the modified VirScan technology, as illustrated in Figure 1 [7]. A phage display library, derived from fragments of N, S, M, and E proteins, and variants of these proteins (downloaded from the National Center for Biotechnology Information on 7 April 2021), was constructed and used for immunoprecipitation with sera from patients hospitalized due to COVID-19 (SARS-CoV-2 variants causing those infections were identified; they all belonged to Alpha or Delta type). After the NGS of the immunoprecipitated libraries, two types of analysis were conducted: (i) the identification of regions frequently recognized in many variants by cross-reacting antibodies (*IgG hotspots*), (ii) the identification of immunogenic regions within structural proteins of SARS-CoV-2.

### 3.1. Some Protein Regions Can Be Altered without Affecting Their Recognition by Patients IgG Antibodies

Protein regions frequently recognized by cross-reacting IgG antibodies in many SARS-CoV-2 variants, despite amino acid differences, were identified as *cross-reacting IgG hotspots* (Figure 2, Appendix A). The library of oligopeptides representing structural SARS-CoV-2 proteins of multiple variants (34,949 oligopeptides, including Omicron-derived sequences, Appendix A) was immunoprecipitated with sera from the investigated patients. The fraction of variants recognized by patients’ IgG was calculated (Figure 2) and normalized to the total number of each oligopeptide variant cloned into the library (this number differed between protein regions; see Appendix A). A high fraction of cross-reacting variants (over 20%, *p* < 0.05) indicates a *hotspot*, while low scores indicate regions where new variants efficiently escaped antibodies induced by Alpha or Delta. The majority of cross-reacting *IgG hotspots* were identified in proteins N and S (Figure 2), while only one was identified in protein M, and none in protein E (Appendix A). High similarity can be observed between the *IgG hotspots* identified by Alpha or Delta-induced IgG, thus suggesting regularities in the existing immunity evasion by the virus.

### 3.2. Immunogenicity Differs between Protein Regions

Further, identification of the immunogenic regions was conducted by detecting homogenous IgG-oligopeptide reactions with the use of a virus-representative phage display library. These reactions represented binding between oligopeptides (in the phage display library) derived from reference SARS-CoV-2 virus and antibodies induced in patients hospitalized due to COVID-19. Fractions of patients whose sera contained IgG targeting defined regions of the proteins N and S are presented in Figure 3. In the majority of cases, immunogenic regions are similar between tested SARS-CoV-2 variants (Figure 3, Appendix A). In proteins M and E, in turn, regions of significant immunogenicity were not detected (Appendix A).

### 3.3. Some Protein Regions Show Both High Immunogenicity and Presence of Cross-Reactive Hotspots

The immunogenic regions (Figure 3, Appendix A) are not fully consistent with the cross-reactivity *IgG hotspot* regions (Figure 2, Appendix A). In protein N, there are two regions that demonstrate both significant immunogenicity and the activity of a cross-reacting *IgG hotspot* located between aa positions 197 and 280 or between 358 and 419. In protein S, four regions that demonstrate both significant immunogenicity and the activity of a cross-reacting *IgG hotspot* were identified: between 281 and 337, 533 and 617, 925 and 1004, and 1145 and 1232. Thus, immunogenic regions with *IgG hotspot* properties are located both in the S1 and S2 domains of protein S. The first and second are located at the C-terminus of the NTD domain and in the RBD domain, respectively. Their comparison and the list are presented in Figure 4 and Table 1. This strongly suggests that they can be useful as virus-neutralizing antibody targets, thus potentially being important in vaccine composition or in monoclonal antibody production. This is due to the increased efficacy of these regions in antibody induction (immunogenicity, and, at the same time, decreased the probability of natural immune evasion of existing immunity against the virus demonstrated by multiple variants of the SARS-CoV-2 virus. The last two immunogenic regions with *IgG hot-spot* properties are located within the HR1 and HR2 domains, which have also been proposed as efficient targets for antibodies neutralizing the virus [14]. The latter extends to a transmembrane domain (TM), the only region within protein S that is hydrophobic [15]. Thus, potentially less accessible for immune response formation. However, this issue requires further studies for full elucidation. 

## 4. Discussion

The potential significance of antibody cross-reactivity is crucial due to the challenge of existing immunity evasion by viruses and other infectious agents. *IgG hotspots* are regions of a protein where the cross-reactivity of antibodies limits the viral potential for evading recognition by the human immune system. Rational strategies for managing expanding pathogenic variants, like those observed in SARS-CoV-2 pandemics, require predictions and efficient estimations of whether emerging variants have a high or limited potential for existing immunity evasion. To some extent, these predictions can be conducted through theoretical comparisons with existing epitope databases (such as those of other viruses), however, experimental verification remains necessary. Given the multitude of emerging variants, assessing pathogens with a high potential for epidemic spread, like SARS-CoV-2, can be challenging for experimental verification of multi-directional cross-reactions. The VirScan method was previously employed to compare the wild-type SARS-CoV-2 virus with other coronaviruses such as MERS, the first SARS, the “common cold” HCov group, and coronaviruses infecting bats [16]. Here, we propose a high-throughput technology that utilizes a phage display library of epitopes derived from a pool of identified viral variants, enabling massive testing of a multitude of epitopes. In this study, 34,949 oligopeptides representing all variants of SARS-CoV-2 available in databases at the moment of the library creation have been tested in a single assay (Appendix A). 

To assess the proposed approach, we focused on two selected proteins in SARS-CoV-2, due to these proteins’ biological and medical importance: nucleocapsid protein N and spike protein S. In the case of the N protein, cross-reactions enhance its suitability as the diagnostic target in serological testing. Consequently, cross-reactivity hotspots represent the regions where new mutations can probably be tolerated, without losing diagnostic applicability. It is, however, less probable (if at all) that antibodies targeting the N protein might have any effect on virus neutralization. As for the S protein, its major significance lies in anti-COVID-19 vaccines and the protective potential of antibodies targeting this protein. Cross-reacting hotspots in this context indicate regions where new mutations of the virus are relatively ‘safe’ for the human population. That is, the probability of existing immunity evasion by a new variant with a mutation in a cross-reactivity hot-spot is lower. Conversely, regions outside the cross-reacting hotspots suggest a higher probability of immunity evasion by a new variant of that type. IgG hotspots identified in this study were located in domains NTD, RBD, HR1, and HR2/TM, which have been recognized as important targets for virus-neutralizing antibodies. 

To identify cross-reactions within protein regions that have overall significant importance for immune responses to the virus, we also assessed the immunogenicity (intensity of region-specific antibody production). Immunogenic regions within proteins N and S of Alpha and Delta variants identified in our work (Figure 3) were similarly located to those identified by VirScan for the wild-type virus previously [16]. This advocates for the robustness of VirScan technology and demonstrates at least some conservancy of immunogenic regions in viral variants.

Of note, this study has focused on epitopes in the majority linear. Specifically, structural epitopes with all amino acids located within the tested 56-amino acid regions could be investigated, while more distant locations rendered detection inefficient. Thus, the potential effect of mutations in many structural epitopes could not be identified, and it may still contribute to specific effects of the immune response and protection. However, even linear epitopes have been demonstrated as important targets in anti-SARS-CoV-2 protection [5,17]. They are also of key importance for applications where a partial protein is used to elicit or detect the immune response. Therefore, we propose the cross-reacting regions of the N and S proteins in SARS-CoV-2 as the major targets in both diagnostics and vaccine design. Of note, the goal of this study was to assess the applicability of the approach (immunoprecipitation of a phage display library presenting viral variants epitopes) in the experimental detection of IgG cross-reactions and the viral evasion of existing immunity against the virus. However, this assessment was completed with exemplary human sera specific to Alpha and Delta variants of SARS-CoV-2. Future studies should include antibodies induced by other viral variants. 

Identified *IgG hotspots* can be helpful in theoretical predictions of the potential spread of newly identified variants, particularly when experimental data regarding a specific variant are not available yet. Variants with new mutations outside the cross-reacting hotspots have the highest potential for immune evasion and, eventually, for spreading in the human population regardless of the vaccination rate or immunity acquired from infections caused by previous variants.

## Figures and Tables

**Figure 1 viruses-16-00058-f001:**
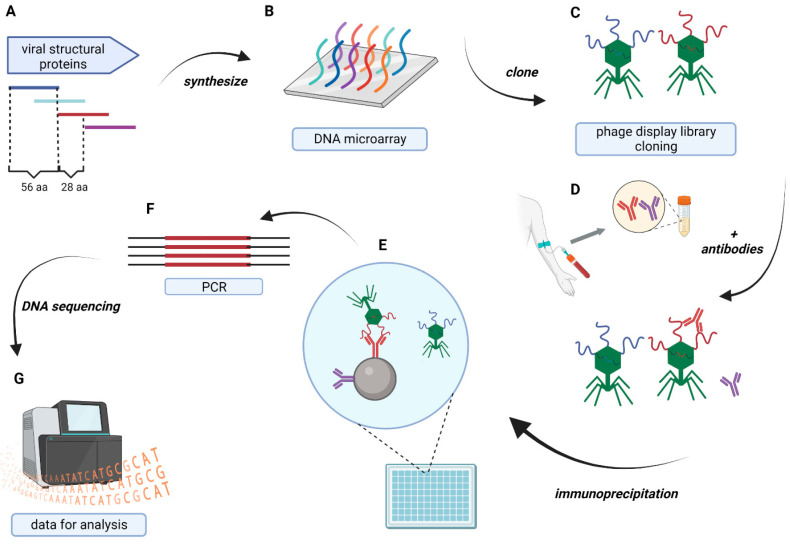
VirScan technology is used for immunoreactivity measurements. (**A**) In silico design of oligopeptides representing chosen proteomes, (**B**) synthesis of oligonucleotides coding for the peptides, (**C**) constructing of phage display library of SARS-CoV-2-derived peptides, (**D**) reaction of the library with patients sera, (**E**) immunoprecipitation with magnetic beads binding Fc fragments of antibodies, (**F**) amplification by PCR, and (**G**) NGS sequencing, modified from Xu et al. [7].

**Figure 2 viruses-16-00058-f002:**
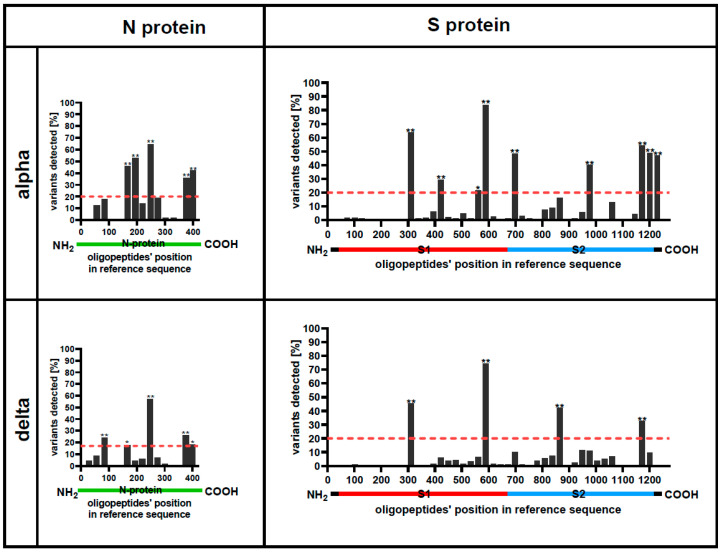
Cross-reacting *IgG hotspots* in the N and S protein of SARS-CoV-2. Cross-reacting IgG hotspots are regions of protein frequently recognized in many variants by cross-reacting antibodies. Cross-reactions of IgG antibodies were identified by the immunoprecipitation of the library containing SARS-CoV-2 variants’ oligopeptides (VirScan technology). Immunoprecipitation was conducted with sera from patients hospitalized due to SARS-CoV-2 infection (Alpha or Delta). The X-axis represents sequentially arranged oligopeptides that sum up to the whole protein sequence, presented from the N-terminus to the C-terminus. Each bar represents the fraction of tested viral variants (see Appendix A) that were recognized by antibodies in patients’ sera (N_alpha_ = 15, N_delta_ = 8). Green lines represent the N protein. Red, blue, and black lines under the plots represent S protein. *—*p* < 0.05, **—*p* < 0.001, one-tail. Red-dotted lines indicate a cut-off *p* < 0.05. *p*-values are calculated between experimental data and a statistical binomial distribution model assuming a random distribution of hotspot regions in proteins.

**Figure 3 viruses-16-00058-f003:**
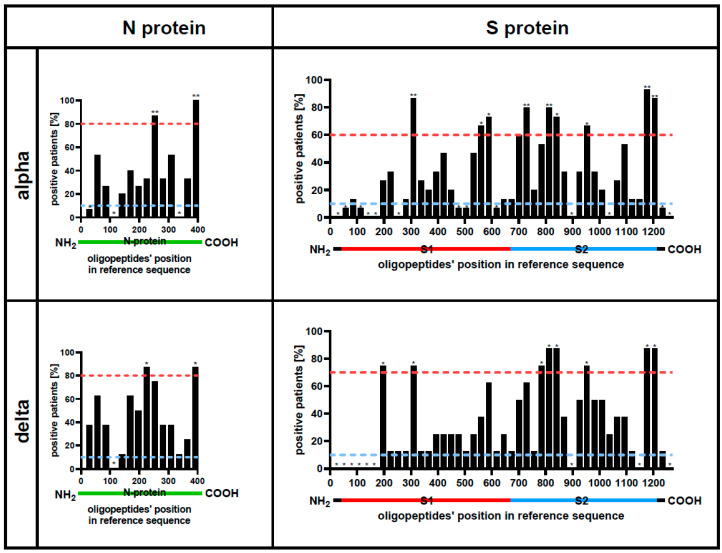
Immunogenic regions of SARS-CoV-2 protein N and S. Immunogenic regions efficiently induce specific IgG production in SARS-CoV-2 infected patients. Immunogenic regions were identified by immunoprecipitation of oligopeptide library representing Alpha and Delta SARS-CoV-2 variants (VirScan technology). Immunoprecipitation was conducted with sera from patients hospitalized due to SARS-CoV-2 infection (Alpha or Delta). The X-axis represents sequentially arranged oligopeptides that sum up to the whole protein sequence, presented from the N-terminus to the C-terminus. Each bar represents the fraction of tested patients who were positive for antibodies specific to a relevant region within SARS-CoV-2 proteins (N_alpha_ = 15, N_delta_ = 8). Green lines under the plot represent the N protein. Red, blue, and black lines under the plot represent the S protein. *—*p* < 0.05, **—*p* < 0.001. Red and blue dotted lines indicate the upper and lower cut-off *p* < 0.05 (two-tail), respectively. *p*-values are calculated between experimental data and the statistical binomial distribution model assuming a random distribution of hot-spot regions in proteins.

**Figure 4 viruses-16-00058-f004:**
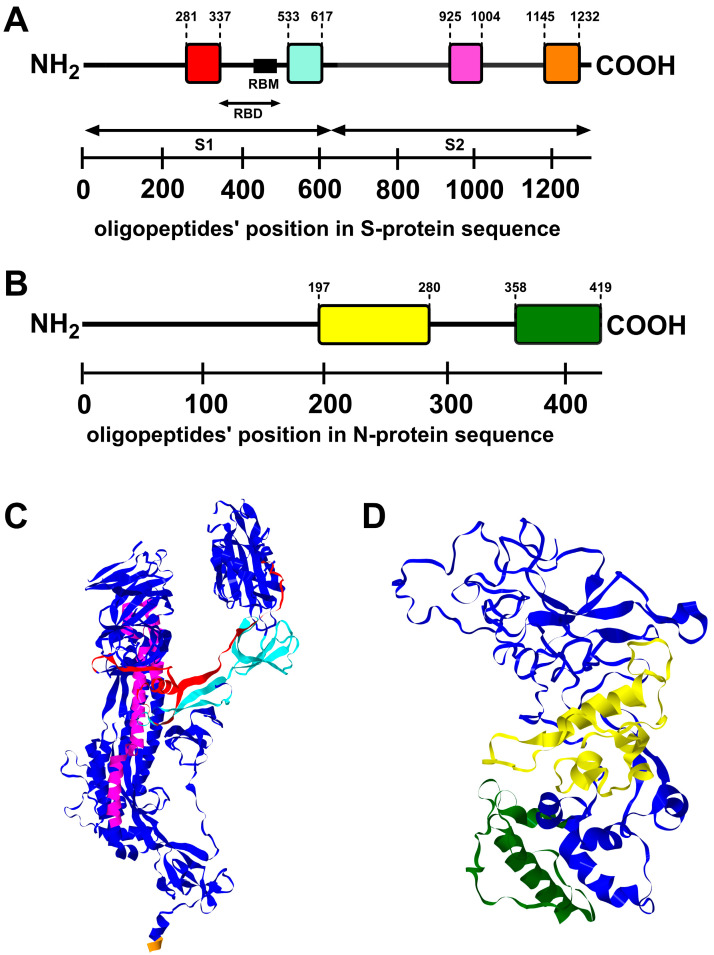
Representation of highly immunogenic regions with strong cross-reacting *IgG hotspots* on sequences and 3D models of proteins S and N. Representation of highly immunogenic regions with strong cross-reacting hotspots (as colored bars) on S protein sequence representation (**A**), N protein sequence representation (**B**), S protein 3D model (PDB ID: 6VXX) (**C**), and N protein 3D model (PDB ID: 8FG2) (**D**). Immunogenic regions efficiently induce specific IgG production in SARS-CoV-2 infected patients, and these were identified by the immunoprecipitation of the oligopeptide library representing Alpha and Delta (N_alpha_ = 15, N_delta_ = 8) SARS-CoV-2 variants (VirScan technology). Cross-reacting IgG hotspots are regions of protein frequently recognized in many variants by cross-reacting antibodies from patients. IgG hotspots were identified by immunoprecipitation of the library containing SARS-CoV-2 variants’ oligopeptides (VirScan technology). Immunoprecipitation was conducted with sera from patients hospitalized due to SARS-CoV-2 infection (Alpha or Delta). Black lines represent an amino acid sequence of proteins. Colored boxes represent regions of respective protein with high or very high immunogenicity (*p* < 0.05 and *p* < 0.001, respectively) and strong cross-reacting hotspots (*p* < 0.001). *p*-values are calculated between experimental data and a statistical binomial distribution model, assuming random distribution. Black arrows represent the S1 and S2 subunits and the RBD of the S protein, and the black box represents the RBM of the S protein. Dark blue colors illustrate the 3D structure of proteins outside the presented hotspots. The number of amino acid residues on the borders of the illustrated fragments is shown at the end of the dotted lines. *p*-values are calculated between experimental data and the statistical binomial distribution model assuming a random distribution of hot-spot regions in proteins.

**Table 1 viruses-16-00058-t001:** Highly immunogenic regions with strong cross-reacting *IgG hotspots* in proteins N and S. Immunogenic regions efficiently induce specific IgG production in SARS-CoV-2 infected patients. Immunogenic regions were identified by the immunoprecipitation of the oligopeptide library representing Alpha and Delta SARS-CoV-2 variants (VirScan technology). Cross-reacting IgG hotspots are regions of protein frequently recognized in many variants by cross-reacting antibodies. IgG antibodies were identified by the immunoprecipitation of the library containing SARS-CoV-2 variants’ oligopeptides (VirScan technology). Immunoprecipitation was conducted with sera from patients hospitalized due to SARS-CoV-2 infection (Alpha or Delta). Listed below are regions of high or very high immunogenicity (*p* < 0.05 and *p* < 0.001, respectively) and strong cross-reacting hotspots (*p* < 0.001). *p*-values are calculated between the experimental data and the statistical binomial distribution model assuming a random distribution of immunogenicity and hot-spot regions in proteins.

	Reference Protein Fragment [Amino Acid Position]		Immunogenicity		Cross-Reacting IgG Hot-Spots	
N-protein	**Start**	**End**	**Alpha**	**Delta**	**Alpha**	**Delta**
	197	280	Very High	High	Strong	Strong
	358	419	Very High	High	Strong	Strong
S-protein	281	337	Very High	High	Strong	Strong
	533	617	High	Medium	Strong	Strong
	925	1004	High	High	Strong	None
	1145	1232	Very High	High	Strong	Strong

## Data Availability

The data underlying this study are not publicly available due to patient privacy issues. The data are available from the corresponding author upon reasonable request (krystyna.dabrowska@hirszfeld.pl).

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
