# Peer review of "Experimental Identification of Cross-Reacting IgG Hotspots to Predict Existing Immunity Evasion of SARS-CoV-2 Variants by a New Biotechnological Application of Phage Display"

_viruses, 2023, doi:10.3390/v16010058_

Round 1

Reviewer 1 Report

Comments and Suggestions for Authors

The authors have applied the VirScan technology to the Apha and Omicron variants of SARS-CoV-2 to identify linear epitopes inside the sequences of the structural proteins that react intensively with serum antibodies in people hospitalized with a diagnosis of COVID-19.

The topic and the presented data are interesting. Yet, I did not find enough novelty in this research. Earlier, a comprehensive paper was published on the same topic: doi: 10.1126/science.abd4250. (Epub 2020 Sep 29.)

Actually, I found two moments of novely: (1) taking the "local" experimental material (Healthcare Centre Bolesławiec, Poland); (2) the bioinformatics way of calculation applied to the experimental data obtained via the (somewhat) modified VirScan protocol. I believe the authors should stress more the first point in the paper and explain better the scientific value of their modifications of the VirScan protocol. It would be nice if the authors discuss and compare their data with the previously published data regarding the SARS-CoV-2 structural proteins (e.g. see the paper mentioned above) .

The authors determine two parameters from the experimental data: (1) "Cross-reacting IgG hot spots" and (2) Immunogenicity. What is the "Scientific Soundness" of their approach compared to the previuos protocols? (Why is it more appropriate than the previous way of describing the linear epitopes?) Actually, both calculation protocols give very similar results.

The second point is: despite there are "linear" epitopes described in this research it would be highly desirable to see those epitopes mapped on the 3D-structures of S and N proteins. It helps visualize them on the proteins' surfaces, that is really important for further vaccine design. Could the authors provide such maps?

Finally, I fould some inconsistency of the data interpretation regarding the S protein amino acid regions, which efficiently "induce specific IgG production" and their attribution to the specific structural elements within the protein. Namely, the authors indicate the aa region 1145 - 1232 as the "immunogenic region with IgG hot-spot properties located within HR2 domain". Yet, according to GenBank Ac. No. QHD43416.1 the HR2 includes amino acids 1162 to 1203, which is follows by a linker and a transmembrane domain (supposedly aa 1212 - 1234). Could the authors comment about the TM domain: it hardly possesses an immunogenic activity, doesn't it?

Author Response

We are most grateful for all comments and advice from the Reviewer, we did our best to address all indicated issues.

Comments 1: The authors have applied the VirScan technology to the Alpha and Omicron variants of SARS-CoV-2 to identify linear epitopes inside the sequences of the structural proteins that react intensively with serum antibodies in people hospitalized with a diagnosis of COVID-19. The topic and the presented data are interesting. Yet, I did not find enough novelty in this research. Earlier, a comprehensive paper was published on the same topic: doi: 10.1126/science.abd4250. (Epub 2020 Sep 29.) Actually, I found two moments of novely: (1) taking the "local" experimental material (Healthcare Centre
Bolesławiec, Poland); (2) the bioinformatics way of calculation applied to the experimental data obtained via the (somewhat) modified VirScan protocol. I believe the authors should stress more the first point in the paper and
explain better the scientific value of their modifications of the VirScan protocol. It would be nice if the authors discuss and compare their data with the previously published data regarding the SARS-CoV-2 structural proteins
(e.g. see the paper mentioned above).
Response 1: Thank you for this important comment. We fully agree that the novelty elements of our study require further clarification and emphasis. In addition to the points raised in this comment, we would like to underscore that our high-throughput analysis also incorporates all recognized SARS-CoV-2 variants at the time of our library design. To elaborate, for the library creation, we downloaded approximately 200,000 variations of sequences; subsequently, homological regions of the investigated proteins were collapsed, resulting in the identification of 96,623 epitopes. Due to this inclusion of all available (at hta moment) variants, our study allowed for identification of cross-reactions of antibodies induced by two specific SARSCoV-2 variants of concern against a diverse array of other variants which were available in the database at the time of our library design.
We believe that this comprehensive analysis represents a major breakthrough compared to the referenced publication (doi: 10.1126/science.abd4250). In the mentioned work, the reference SARS-CoV-2 virus (first sequence) was primarily compared to other coronaviruses such as MERS, the first SARS, the
"common cold" HCov group, and coronaviruses infecting bats. Consequently, we propose an extended discussion covering both the points raised by the Reviewer and those presented herein, underscoring the significance of our approach in advancing the understanding of SARS-CoV-2 variants and their potential cross-reactivity.

Comments 2: The authors determine two parameters from the experimental data: (1) "Cross-reacting IgG hot spots" and (2) Immunogenicity. What is the "Scientific Soundness" of their approach compared to the previous protocols? (Why is it more appropriate than the previous way of describing the linear epitopes?) Actually, both calculation protocols give very similar results.
Response 2: We affirm the validity of identifying immunogenicity regions in both our current protocol and previous methodologies, as it constitutes the core application of all VirScan-derived assays. By extending the scope of the library, experimental assay, and calculations to encompass multivariant analysis, we have introduced a powerful new tool for predictions grounded in empirically observed interactions. Consequently, we propose a combined approach involving the identification of immunogenicity regions and the pinpointing of cross-reacting IgG hot spots. This synergistic strategy promises to yield more valuable insights into the immunogenicity of the virus (or other pathogens). We propose more elucidation of this point within the manuscript, hoping that the extended explanation will meet the Reviewer's expectations.

Comments 3: The second point is: despite there are "linear" epitopes described in this research it would be highly desirable to see those epitopes mapped on the 3D-structures of S and N proteins. It helps visualize them on the proteins' surfaces, that is really important for further vaccine design. Could the authors provide such maps?

Response 3:Thank you for this suggestion; it is similar to other Reviewers’ comments, so aiming to address them all we propose an additional panel in Figure 4, with 3D presentations of the regions of interest within the
protein structures.

Comments 4: Finally, I found some inconsistency of the data interpretation regarding the S protein amino acid regions, which efficiently "induce specific IgG production" and their attribution to the specific structural elements within the protein. Namely, the authors indicate the aa region 1145 - 1232 as the "immunogenic region with IgG hot-spot properties located within HR2 domain". Yet, according to GenBank Ac. No. QHD43416.1 the HR2 includes amino acids 1162 to 1203, which is follows by a linker and a transmembrane domain (supposedly aa 1212 - 1234). Could the authors comment about the TM domain: it hardly possesses an immunogenic activity, doesn't it?
Response 4:Thank you for this comment, we have corrected information on the domains covered by the identified region accordingly. We agree that TM is less probable to be highly immunogenic, however we believe that this issue requires further studies, so we propose it as an additional comment in the discussion.

Reviewer 2 Report

Comments and Suggestions for Authors

I thank the authors for the original and interesting article. The authors proposed a method based on the modified VirScan technology. Here, protein hotspots of various virus variants to which IgG cross-react are identified. Identified IgG hot-spots can be helpful in theoretical predictions of the potential for spread of newly identified variants, particularly when experimental data regarding a specific variant are not available yet.

There are minor issues that must be corrected prior to Acceptance. Listed below.

1. Introduction Section: The statement “N protein, in turn, seems to be a key target for serological diagnostics” is not true, since most serological tests target of the S protein.

2. “However, one cannot exclude a role of these antibodies in virus inhibition, for instance via complement system activation or facilitating phagocytosis.” This statement is questionable. Please provide supporting articles.

3. “After two years of the pandemic and virus evolution within the human population, it is almost impossible to explain all possible combinations of previous infections with different variants, a few types of vaccinations, and new variants that may infect an individual patient.” This sentence is not clear.

4. Section 2.1. Please indicate the sample collection period.

5. Section 2.3. “then stored at – 20°C for further use”. The word “serum” is missing here.

6. Section 2.4. There is no information on the collection of nasopharyngeal swabs.

7. Section 2.4. Please provide the sequences of the primers used?

8. Section 2.4. “software provided by the manufacturer”. The manufacturer should be cited here.

9. Section 2.5. “single and triple alanine substitution, similarly to original procedure.” What is the point of this procedure? Please inquire about replacements.

10.  Section  3. Please indicate the size of the resulting phage display library.

11.  Section 3.1. Please indicate all virus variants used and the number of sequences of each variant.

Author Response

We are most grateful for all comments and advice from the Reviewer, we did our best to address all indicated issues.

Comments 1: I thank the authors for the original and interesting article. The authors proposed a method based on the modified VirScan technology. Here, protein hotspots of various virus variants to which IgG cross-react are identified. Identified IgG hot-spots can be helpful in theoretical predictions of the potential for spread of newly identified variants, particularly when experimental data regarding a specific variant are not available yet. There are minor issues that must be corrected prior to Acceptance. Listed below. 1. Introduction Section: The statement “N protein, in turn, seems to be a key target for serological diagnostics” is not true, since most serological tests target of the S protein.

Response 1: Thank you for drawing our attention to this, we fully agree and we correct the statement.

Comments 2: “However, one cannot exclude a role of these antibodies in virus inhibition, for instance via complement system activation or facilitating phagocytosis.” This statement is questionable. Please provide supporting articles.

Response 2: Thank you for this comment, indeed the effects of complement system activation in COVID-19 are complex, often contradictory findings have been presented. In addition to protective effects, complement system may contribute to cytokine storm activation and to severity of the disease. Thus,
we remove this questionable fragment.

Comments 3: “After two years of the pandemic and virus evolution within the human population, it is almost impossible to explain all possible combinations of previous infections with different variants, a few types of vaccinations, and
new variants that may infect an individual patient.” This sentence is not clear.

Response 3: We propose a simplified version: “After two years of the pandemic and the continuous emergence of new variants, it is challenging to thoroughly understand all the possible combinations of previous infections, various vaccination types, and potential exposure to new variants in an individual patient”,we hope it will be found more clear.

Comments 4: Section 2.1. Please indicate the sample collection period.

Response 4: We have added information as recommended.

Comments 5: Section 2.3. “then stored at – 20°C for further use”. The word “serum” is missing here.

Response 5: We have added information as recommended.

Comments 6: Section 2.4. There is no information on the collection of nasopharyngeal swabs.

Response 6: Thank you for indicating this missing information, we have added as recommended.

Comments 7: Section 2.4. Please provide the sequences of the primers used?

Response 7: We have added information as recommended.

Comments 8: Section 2.4. “software provided by the manufacturer”. The manufacturer should be cited here.

Response 8: We have added information as recommended.

Comments 9: Section 2.5. “single and triple alanine substitution, similarly to original procedure.” What is the point of this procedure? Please inquire about replacements.

Response 9:Thank you for indicating this unclear element. We propose following additional explanation for this step in the procedure: “Alanine substitutions have been established in VirScan technology as the method for
epitope disruption. Specifically, alanine is substituted for each amino acid in the investigated sequence of the epitope, both individually and in triplicates. In the case of original sequences containing alanines, the substitution is completed with glycine. These substitutions are employed to identify specific amino acids involved in epitope-antibody interactions. Disruption of the interaction occurs when a key amino acid is substituted.

Comments 10: Section 3. Please indicate the size of the resulting phage display library.

Response 10: We have added information as recommended.

Comments 11: Section 3.1. Please indicate all virus variants used and the number of sequences of each variant.

Response 11: Regrettably, it is not feasible for us to enumerate all variants, as we have utilized every variant accessible in the database. These variants consist of individually uploaded genomes, frequently lacking specific
variant names. Post-download, all homologous sequences were condensed (duplications were removed) to prevent the inclusion of redundant sequences in the library. Therefore, a single 56-amino acid sequence in the library may denote multiple viral variants, provided they exhibit identical molecular
features, and providing the number of sequences for each variant is not possible. This methodology is a product of the VirScan technology. To address the variants issue, we provide a specific date for their data downloading and we specifically list all sequences used in the library (Supplementary). We trust that this elucidation meets the Reviewer's understanding.

Reviewer 3 Report

Comments and Suggestions for Authors

The paper is dedicated to identification of the SARC-CoV-2 N- and S-protein regions frequently recognized by cross-reacting IgGs developed after different versions of the infection, using phage display. To my mind, the study is very interesting and useful for current studies and future applications.

However, the paper must be improved prior to publication

  1. English needs editing (except for the abstract)

  2. In the abstract, please describe your research in more detail. Now it looks more like a part of an introduction. 

  3. Introduction - “N protein, in turn, seems to be a key target for serological diagnostics. This relatively conserved protein has commonly been used in serological diagnostics, particularly to confirm SARS-CoV-2-related etiology of already cleared infections,since N protein is unique for SARS-CoV-2, highly immunogenic, and not included in anti-COVID-19 vaccines [6].” - this is not true. (a) IgG antibodies to the N-protein may not be developed in the case of light or asymptomatic course of the infection; (b) there are vaccines made on the basis of inactivated virus (like sinopharm) that definitely contain the N-protein. Also, as far as I know, now the peptide vaccines are being developed which contain different fragments of the virus, including the fragments of N-protein. 

  4. “Here we validate a new method for experimental identification of regions (linear epitopes) that can play a role of cross-reacting IgG hot-spots in SARS-CoV-2 structural proteins.” - it is better to specify that N and S were studied.

  5. Please, describe the methodology of genotyping in more detail - what primers did you use? How the PCR and libraries prep was set up, etc. Now the description of this part is too brief. 

  6. All figures seem to be duplicated in the current version

  7. Figure 4 - it will be more useful if you mark the numbers of aminoacid residues on the borders of immunogenic sequences. Also, the RBD and the RBM borders can also be marked. Ideally, positioning of the cross-reactive regions can be shown on the protein structure. 

  8. In the discussion section, please compare your findings with the other studies of the conservative/cross-reacting regions - how do they match? Are your regions conservative in different strains or not?

Comments on the Quality of English Language

English needs editing

Author Response

We are most grateful for all comments and advice from the Reviewer, we did our best to address all indicated issues.

Comments 1: The paper is dedicated to identification of the SARC-CoV-2 N- and S-protein regions frequently recognized by cross-reacting IgGs developed after different versions of the infection, using phage display. To my mind, the study is very interesting and useful for current studies and future applications. However, the paper must be improved prior to publication
1. English needs editing (except for the abstract)

Response 1: The manuscript has been corrected, we hope it will be found improved now.

Comments 2: In the abstract, please describe your research in more detail. Now it looks more like a part of an introduction.

Response 2: Thank you for this comment, we introduced some additional method and results description. This additional information is not extensive due to important limitations in the Abstract length, but we hope our extension will be found sufficient.

Comments 3: Introduction - “N protein, in turn, seems to be a key target for serological diagnostics. This relatively conserved protein has commonly been used in serological diagnostics, particularly to confirm SARS-CoV-2-related etiology of already cleared infections,since N protein is unique for SARS-CoV-2, highly immunogenic, and not included in anti-COVID-19 vaccines [6].” - this is not true. (a) IgG antibodies to the N-protein may not be developed in the case of light or asymptomatic course of the infection; (b) there are vaccines made on the basis of inactivated virus (like sinopharm) that definitely contain the N-protein. Also, as far as I know, now the peptide vaccines are being developed which contain different fragments of the virus, including the fragments of N-protein.

Response 3: Thank you for drawing our attention to this problem. The comment is similar to a comment given by Reviewer 2. This part has been modified accordingly.

Comments 4: “Here we validate a new method for experimental identification of regions (linear epitopes) that can play a role of cross-reacting IgG hot-spots in SARS-CoV-2 structural proteins.” - it is better to specify that N and S were studied.

Response 4: Our most interesting observations refer to proteins S and N, however proteins E and M have also been tested (with no significant identifications). Thus we propose to keep “structural proteins” in this sentence, but we correct the following statement to make this part more clear and relevant.

Comments 5: Please, describe the methodology of genotyping in more detail - what primers did you use? How the PCR and libraries prep was set up, etc. Now the description of this part is too brief.

Response 5: Thank you for this comment, this description has been extended as recommended. More details have been added in Section 2.5: “Epitope analysis”.

Comments 6: All figures seem to be duplicated in the current version.

Response 6: We apologize for this problem, it results from the properties of uploading system, it can be solved at the step of further manuscript processing (if accepted).

Comments 7: Figure 4 - it will be more useful if you mark the numbers of amino acid residues on the borders of immunogenic sequences. Also, the RBD and the RBM borders can also be marked. Ideally, positioning of the cross-reactive regions can be shown on the protein structure.

Response 7: Thank you for this comment, the advice to use protein structure is similar to that given by Reviewer 1. We propose a new version of Figure 4 with additional panels presenting regions identified in this study in the proteins’ structures.

Comments 8: In the discussion section, please compare your findings with the other studies of the conservative/cross-reacting regions - how do they match? Are your regions conservative in different strains or not?

Response 8: We propose comparison of our work to other VirScan technology-based study. Immunogenic regions identified in that one were similar, even though the study was done with wild-type virus (while our with Alpha and Delta variants). However cross-reactions in a high throughput assay were only identified with other coronaviruses, like MERS, so called ‘common cold’ coronaviruses, or bat-infecting coronaviruses, our multivariant identification of IgG hot spots seem unique. We propose and extension within Discussion section.

Reviewer 4 Report

Comments and Suggestions for Authors

In their manuscript, Harhala describe a phage display assay to map crossreactive epitopes in SARS-CoV-2 variants. This is an interesting approach and the manuscript should be published provided a few points are addressed.

1.      General: the authors use the term immune evasion throughout the whole manuscript. However, immune evasion means an active inhibition of an immune response like the interruption of a signaling cascade. What the authors mean is the evasion of existing immunity against the virus. They should correct this term.

2.      Abstract: racing is certainly the wrong expression here.

3.      Results paragraph 3.2: it should be stated at the beginning that the authors used also here phage display. The way it is written seems to indicate that peptides were tested.

4.      Results table 1: why are some words bold and others not. Explain.

5.      Discussion: the authors claim that the epitopes they test are linear. However, petide of the length they express might have a correct secondary structure. It should be expressed more carefully.

6.      Results/Discussion: what is missing to my mind is to indicate were epitopes are found which have been proven be detected by neutralizing antibodies.

Author Response

We are most grateful for all comments and advice from the Reviewer, we did our best to address all indicated issues.

In their manuscript, Harhala describe a phage display assay to map cross-reactive epitopes in SARS-CoV-2 variants. This is an interesting approach and the manuscript should be published provided a few points are addressed.

Comments 1: General: the authors use the term immune evasion throughout the whole manuscript. However, immune evasion means an active inhibition of an immune response like the interruption of a signaling cascade. What the authors mean is the evasion of existing immunity against the virus. They should correct this term.

Response 1: Thank you for drawing our attention to this issue, we have now corrected this expression throughout the manuscript, including proposition for the title modification.

Comments 2: Abstract: racing is certainly the wrong expression here.

Response 2: We have corrected the indicated sentence.

Comments 3: Results paragraph 3.2: it should be stated at the beginning that the authors used also here phage display. The way it is written seems to indicate that peptides were tested.

Response 3: Thank you for this advice, we have corrected the text accordingly.

Comments 4: Results table 1: why are some words bold and others not. Explain.

Response 4: Thank you for this comment, we have corrected formatting in the table (bold removed).

Comments 5: Discussion: the authors claim that the epitopes they test are linear. However, peptide of the length they express might have a correct secondary structure. It should be expressed more carefully.

Response 5: Thank you for this very interesting comment, indeed our phage-displayed oligos may present secondary structure, thus offering more than linear epitope representation only. We did our best efforts to give relevant explanation on that, we hope it will be found adequate. 

Comments 6: Results/Discussion: what is missing to my mind is to indicate were epitopes are found which have been proven be detected by neutralizing antibodies.

Response 6: Thank you for this comment. We have added information on potentially neutralizing antibodies targets within the proteins, we also add more structural information by extending Figure 4 with new panels. We hope this will be found helpful and sufficient.

Round 2

Reviewer 1 Report

Comments and Suggestions for Authors

The authors have improved the paper essentially.

I am satisfied with all the explanations.